# *Urolithiasis* as a Husbandry Risk to Yaks in the Swiss Alps

**DOI:** 10.3390/ani14192915

**Published:** 2024-10-09

**Authors:** Michael Hässig, Natascha Biner, Christian Gerspach, Hubertus Hertzberg, Michaela Kühni, Claude Schelling, Annette Liesegang

**Affiliations:** 1Department of Farm Animals, University of Zurich, Winterthurerstrasse 270, CH-8057 Zurich, Switzerland; nawunder@gmx.ch (N.B.); cgerspach@vetclinics.uzh.ch (C.G.); 2Institute of Parasitology, University of Zurich, Winterthurerstrasse 270, CH-8057 Zurich, Switzerland; hubertus.hertzberg@access.uzh.ch (H.H.); michaela_kuehni@bluewin.ch (M.K.); 3Research Platform AgroVet-Strickhof, University of Zurich, Eschikon 27, CH-8315 Lindau, Switzerland; claude.schelling@uzh.ch; 4Institute of Animal Nutrition and Dietetics, University of Zurich, Winterthurerstrasse 270, CH-8057 Zurich, Switzerland; aliese@nutrivet.uzh.ch

**Keywords:** yak, hypercalcemia, *urolithiasis*

## Abstract

**Simple Summary:**

From 2006 to 2014, 10 cases of *urolithiasis* in yaks (*Bos grunniens*) with calcium carbonate uroliths were confirmed in Switzerland. From six problem and four control farms, distributed within different regions in Switzerland, a total of 99 animals were examined. In addition, roughage, soil, and water samples were analyzed. This study revealed different Ca:P ratios, from 1.56 to 7.74:1, in the forages and mild *hypercalcemia* in the animals of the problem farms. In a univariate analysis of the problem versus control farms, about 20 other significant factors emerged. The multivariate analysis suggested that altitude, crude protein (CP), P, Mg, neutral detergent fiber (NDF), and acid-resistant detergent fiber (ADF) in the diet were important factors. Based on analysis of the pedigree, there was no evidence of an obvious genetic background of *urolithiasis*. *Urolithiasis* in yaks in alpine Switzerland seems to be a husbandry risk. Calcium-rich forages play a crucial role in this multifactorial process.

**Abstract:**

Background: Between 2006 and 2014, 10 cases of *urolithiasis* in yaks with calcium carbonate uroliths were confirmed in Switzerland, and at the same time, a sixfold calcium overhang in roughage in an affected farm was evident. The purpose of this study was the question of whether *urolithiasis* in yaks in the alpine regions of Switzerland poses a husbandry risk. The hypothesis was that elevated calcium levels in roughage led to *hypercalcemia* and, thus, the formation of calcium carbonate stones. Methods: Blood samples from 99 animals from 10 farms were examined (*n* = 6 problem farms; n = 4 control farms). Several metabolites were analyzed in the blood and urine. In addition, roughage, soil, and water samples were analyzed. The farms were distributed within different regions of Switzerland. Results: This study revealed different Ca:P ratios, from 1.56 to 7.74:1, in the forages and mild hypercalcemia in the animals of the problem farms. In a univariate analysis of the problem versus control farms, about 20 other significant factors emerged. Multivariate analysis showed that altitude, CP (crude protein), P, Mg, NDF, and ADF in the diet are important factors. Based on analysis of the pedigree, there was no evidence of an obvious genetic background of *urolithiasis*. Limitations: A limitation of this study is the small number of yaks in Switzerland. Conclusions: The question of whether *urolithiasis* in yaks in alpine Switzerland poses a husbandry risk can be answered affirmatively. Calcium-rich forages play a crucial role in this multifactorial process.

## 1. Introduction

Yak farming is becoming increasingly important in Switzerland [1]. Yaks can replace cows at greater elevations. Yaks can graze on the abandoned pastures of both cattle and sheep. In addition, wolves (*Canis lupus*) do not normally harm yaks. Since up to 1 km^2^ of pastureland is now available per yak, depending on the alp, they are often overfed. In the Valais, traditional Eringer cows have become heavier in order to perform better in traditional cow fights. Cows of other breeds are allowed to graze at up to 2000 m a.s.l. Eringer cows currently graze at up to 2400 m a.s.l. In the past, these cows grazed at vegetation levels of even up to 3000 m a.s.l.; however, this is no longer possible due to the heaviness of Eringer cows [1,2]. Consequently, pastures between 2400 and 3000 m a.s.l. have been abandoned, resulting in increased avalanche danger for villages and road traffic in Valais during winter. Thus, yak farming represents ecologically sensible animal husbandry; however, overfeeding in the summertime might be a problem [1].

Yaks (*Bos grunniens*) are Asian wild and domestic cattle. They are classified as ruminants (*Ruminantia*) and horn-bearers (*Bovidae*) belonging to the subfamily *Bovinae* of the genus *Bos*. Very cold winters and cool summers (hardly above a 10 to 15 °C daily maximum temperature) determine the vegetation in Tibet, where yaks originate [1,3]. Yaks are mainly kept for breeding, meat production, and trekking and as hobby animals used for extensive landscape maintenance. In 2021, there were 75 yak keepers, with around 1000 animals present in Switzerland [1].

The pathogenesis of *urolithiasis* has not been described much. The formation of kidney stones is a multifactorial process. Current theories assume a nidus-associated stone formation of calcium-based *urolithiasis*, whereby an unfavorable promoter-to-inhibitor ratio relating to lithogenesis also plays an important role alongside that of pH. In addition, there are also indications of genetic and gender components [4,5,6]. Single genes that increase the risk of stone formation have been identified in dogs [7] and cows [8]. *Urolithiasis* can also be caused by inflammatory processes. Desquamated renal tubule cells act as cores for uroliths. We found no specific scientific work focused on *urolithiasis* in yaks.

Hereditary calcium oxalate *urolithiasis* type 1 (CaOx1) and type 2 (CaOx2) are two different autosomal-recessive genetic disorders that greatly increase the risk of CaOx stone formation in the urinary tract. Urinary drainage disorders are much more common in male domestic ruminants. This fact can be explained by the special anatomy of the *urethra* in male ruminants. The predisposed anatomic locations for urinary obstructions in males are the *sciatic arcus* and the *flexura sigmoidea*.

The Department of Farm Animals, University of Zurich, Switzerland, was asked to help in 10 cases of yak *urolithiasis* between 2006 and 2014. Calcium carbonate uroliths were found, and a sixfold calcium overhang in roughage in the affected farm was observed. In this study, the question of whether *urolithiasis* in yaks in the alpine regions of Switzerland poses a husbandry risk was investigated. The hypothesis was that elevated calcium levels in roughage lead to *hypercalcemia* and, thus, the formation of calcium carbonate stones.

## 2. Materials and Methods

### 2.1. Study Farms

Ten farms were visited between the beginning of December 2016 and mid-April 2017, and roughage, water, soil, urine, fecal, and blood samples were examined. Since there are only about 70–80 farms with a total of only about 1000 animals in Switzerland, it was not possible to randomize farm selection. One case of *urolithiasis* within a farm was considered the criterion for a case farm. Initially, 4 problem farms, i.e., farms in which *urolithiasis* had been detected within the last 11 years (cases), and 6 farms where *urolithiasis* had not been detected over the last 11 years were included in this study (controls). During this study, two farms switched from control farms to problem farms (6 problem farms and 4 control farms) because of extended anamnesis, as new cases of *urolithiasis* appeared during the study period.

The problem farms were exclusively located in mountain zones III (farms at high altitudes and other factors) and IV (farms at very high altitudes and other factors), and the control farms were all located in cadastral zones. Cadastral zones are defined by the Swiss Federal Office of Agriculture, and the relevant criteria include climate, traffic situation, altitude, and exposure. The most important criterion is altitude. The selected farms were located at altitudes between 488 and 1777 m a.s.l. The altitude of each farm is given.

### 2.2. Problem Cases and Sampling

The complaints raised by the farmers, relating to yaks, were severe colic, depression, and anorexia. Seven stones were collected in the slaughterhouse or during gross necropsy. These stones were analyzed at the Laboratory of the Department of Farm Animals, University of Zurich. Since the problem cases all occurred during winter feeding, sampling was also conducted during this period. This study defines winter feeding as not going to pasture and eating roughage and instead eating grass with varying quantity and quality. Additionally, drinking water is more available in summer than in winter in the study region because the water sources freeze in winter. Whether or not minerals were supplied in the right amounts could not be sufficiently evaluated. In some farms, winter feeding started at the beginning of October and lasted until the beginning of June (33 weeks). Except for one farm, which imported part of its basic feed component from Germany (a control farm), all of them used their own or regional roughage for feed.

### 2.3. Blood and Urine Sampling

The aim was to sample 10 animals per farm, ideally including males and females of all ages. At the time of the farm visits, there were 361 animals on 10 farms, with 150 animals on the largest and 9 on the smallest. Blood samples were collected from 94 animals and examined. A sterile Vacutainer^®^ (Becton-Dicinson AG, Allschwil, Switzerland; 10 mL; 367896) containing potassium oxalate (368921), sodium fluoride (2 mL; 367764), and potassium EDTA (5/10 mL; 366643) was used for blood sampling. Very uncooperative animals were excluded from the investigations for safety reasons for both humans and animals. The youngest animal was 14 months old at the time of the blood sampling, and the oldest was 188 months. The median age was 45 months.

Urine could only be collected from 6 animals by means of spontaneous urine. Catheterization for urine collection and ultrasound examinations of the kidneys were not established due to the excessive use of force relative to the value of such information. Sedation of the yaks was not in accordance with Swiss animal welfare law permissions.

Urine analysis was performed on a Cobas Mira S^®^ (Roche, Basel, Switzerland) in the laboratory of the Department of Farm Animals, Department of Outpatient and Population Medicine, Vetsuisse Faculty of the University of Zurich. The parameters used are given in Table 1. Blood samples used to collect serum or plasma were centrifuged and pipetted on site within 6 h after sampling and then sent by express mail with a cold bag to the laboratory of the Department of Farm Animals, Vetsuisse Faculty of the University of Zurich. Serum or plasma tests were also carried out using a Cobas Mira S^®^ (Roche, Basel, Switzerland), in the aforementioned laboratory, with the parameters indicated in Table 1. The Ca, P, protein, pH, and sediment in the urine were not determined in this analysis.

The following investigations were carried out using SYNLAB. Vet. GmbH, Augsburg, Germany provided Hb (hemoglobin) and GSH-Px (glutathione peroxidase). The following investigations were carried out at the laboratory of the Institute of Animal Nutrition and Dietetics, Vetsuisse Faculty University of Zurich (www.tierer.uzh.ch; accessed on 3 October 2024). The laboratory kits were used to evaluate bovine OC (osteocalcin), PTH (parathormon), and 25-hydroxy-Vitamin-D.

### 2.4. Fecal Samples

During the farm visits, fecal samples were obtained rectally. The farmers were instructed not to deworm their animals before this study. The period set prior to the last anthelmintic treatment was at least 6 months. The fecal material was stored at 4 °C and examined within one week after sampling, with the exception of the Baermann technique [9], which was carried out on the same or the following day. The samples were additionally analyzed using the combined sedimentation/flotation technique, the McMaster method (sensitivity of 50 eggs per gram [epg]), and Ziehl–Neelsen staining [9,10]. Fecal cultures for the differentiation of strongyle larvae were conducted using the feces of a maximum of 6 animals each (MAFF, 1986) [9].

### 2.5. Genomic Analysis

The genomic DNA of all yaks was isolated using the Qiagen DNeasy Blood & Tissue Kit (Qiagen 69504, Hilden, Germany). Pedigree-Viewer 6.5 software was used to prepare a pedigree of the yak family [11].

### 2.6. Roughage and Hay Samples

On average, two to three roughage or hay samples per farm were collected during the farm visits, depending on the number of stocks. The values were averaged per farm. They were analyzed at the laboratory of the Institute of Animal Nutrition and Dietetics, Vetsuisse Faculty University of Zurich (http://www.nutrivet.uzh.ch; accessed on 3 October 2024), and sensory, microscopic, and botanical investigations were performed. Furthermore, proximate and van Soest fiber analyses were carried out.

### 2.7. Soil Samples

The minerals and trace elements were analyzed at UFAG Laboratorien AG, Sursee, Switzerland. A detailed description of the methods used can be found at www.ufag-laboratorien.ch (accessed on 17 September 2023). In order to compare the farms, the mean values of the individual farms were used.

Soil samples were taken by the yak farmers themselves at several locations on the roughage pastures. Soil analyses were carried out at the Laboratory for Soil and Environmental Analysis, Thun.

The following soil parameters were analyzed according to the method described by lbu (Labor für Bodenanalyse und Umweltanalytik, Thun, Switzerland), www.lbu.ch (accessed on 17 September 2023):Humus, clay, and silt (touch test);pH (1:2.5 H_2_O);Available (water-soluble) nutrients: nitrate, phosphorus, potassium, calcium, and magnesium;Reserve nutrients: phosphorus, potassium, calcium, and magnesium.

### 2.8. Water Samples

Water samples were taken by the yak farmers themselves. The samples were microbiologically and physico-chemically analyzed at the laboratory of the Food Safety and Veterinary Office of the Canton of Fribourg, www.fr.ch/lsvw (accessed on 8 October 2024).

### 2.9. Statistical Analysis

All of the data were collected in an Excel^®^ spreadsheet and transferred in STATA (StataCorp., 2017; Stata Statistical Software^®^: Release 15.1; College Station, TX, USA: StataCorp LP). All of the data were checked for normal distribution using the Shapiro–Wilk test. Non-normally distributed data were transformed where possible. Normally distributed data were given as means ± standard deviation, and non-normally distributed data were given as medians, minimums, and maximums.

Significant and tendentious differences in the continuous data were tested using analysis of variance (ANOVA), linear regression, and one-way ANOVA with Bonferroni post hoc testing. In the univariate models, an unpaired two-sided *t*-test or a paired two-sided *t*-test was performed for continuous data. For categorical data, the chi-square test was applied at *n* ≥ 5/cell and Fisher’s exact test at *n* < 5/cell. In addition, the “General Linear Model” (GLM) was applied. The Akaike information criterion (AIC) was used to optimize the models. The inclusion criterion for the step-back procedure was a *p*-value of <0.2. A *p*-value of ≤0.05 was considered for the final model (Altmann et al., 1994) [12]. The printout of the final model in STATA with model specifications is given in Appendix A.

A *p*-value of ≤ 0.05 was considered significant, and a *p*-value of 0.05 < *p* < 0.2 was considered to indicate a tendency.

## 3. Results

### 3.1. Differences between Problem and Control Farms

The seven stones that could be examined were calcium carbonate *(n* = 5) or calcite stones (*n* = 2).

The farm locations are shown in Table 2 and Figure 1. There is a significant correlation between m a.s.l. and problem and control farms (*p* < 0.05).

The duration of problems relating to *urolithiasis* at the beginning of this study varied between one and eleven years. The farmers had knowledge of *urolithiasis* problems in their herds. In the cases of two farms, problems only arose after samples were collected.

The average duration of winter feeding in the problem farms (28.0 ± 5.0 weeks) was, on average, seven weeks longer than that in the control farms (20.5 ± 1.7 weeks), with three of the problem farms having relatively short winter feeding periods (not significant).

### 3.2. Blood and Urine Samples

Urine samples from six yaks were examined. Of these, five were taken from problem farms and one was taken from a control farm. All of the values are within the reference range for cattle (Table 3).

Three problem farms have significantly higher average blood levels of Ca. In the case of P, one problem farm has a significantly higher mean value than the reference value for cattle. Two control farms and one problem farm are at the upper limit of the reference value (Table 4). Except for one control farm, all farms have significantly higher values for Mg than the reference values for cattle.

For BHB, all concentrations at the farm level are within the reference range for cattle. However, the values are significantly higher for the problem farms than for the control farms. On the other hand, all values for FFA are above the reference value for cattle. Overall, the values of the control farms are significantly higher.

The values for GGT are increased for two control farms, and the values for GLDH are increased for all farms. The activity of GLDH is significantly higher on the control farms than on the problem farms. It is worth mentioning that one of the control farms had a known problem with *Fasciola hepatica*. As expected, the values obtained from this farm were well above normal values.

### 3.3. Feed Samples

For the feed samples, two to three samples were analyzed per farm. On the control farms, only grass-rich samples were found. The dry matter (DM; *p* < 0.01), CF (crude fiber), NDF (neutral detergent fiber), and ADF (acid-resistant detergent fiber; *p* < 0.001) have significantly higher mean values than those found on the problem farms. The problem farms, on the other hand, show significantly (*p* < 0.001) higher mean values for CP (crude protein), SF (Soxhlet fat), and ADL (acid detergent lignin; Table 5). If the Ca values of roughage are compared at the farm level, only one control farm that obtains its roughage from abroad is within the reference range. Two control farms have values that are too low, and all problem farms are higher than the reference range. The Ca values found in the feed are clearly dependent on the altitude (m a.s.l.) of the farm. In the case of the P values, both the mean value of the problem farms and that of the control farms are below the reference value (Table 4). In this study, the control farms show slightly significantly higher values than the problem farms. On our farms, there was no correlation between the P value in the feed and altitude. Figure 2 shows the high Ca:P ratios of the problem farms as a result of the high Ca and low P values. All the values for K in roughage are within the reference range, which is higher in Switzerland than in other countries. On the other hand, the Na values are well below the reference range for all farms, which is a frequent finding in Switzerland. In the case of S, only one problem farm is within the reference range; the others are all too low. For Mn, Cu, Zn, Se, Co, J, Fe, and Mo, there are large differences between the farms; however, there are no differences between the case and control farms.

### 3.4. Soil Samples

In the soil samples, the differences in humus, clay, silt, soil pH, available N, available P, available K, available Ca, available Mg, reserve P, reserve K, reserve Ca, reserve Mg, soil Bo, soil Mn, soil Cu, and soil Fe between farms were evaluated; however, no differences between the case and control farms were found.

### 3.5. Water Samples

For the mineral content and the degree of hardness of water sources, the control farms show significantly higher values than the case farms (Table 6). The water was softer in the control farms than in the problem farms.

### 3.6. Multivariate and Genomic Analysis

The step-back procedure, taking into account the AIC (Akaike information criterion) and the multivariate model (GLM: General Linear Model) with Gaussian variance function, showed that the altitude (MüM), crude protein (RP_g_100g_TS), NDF (NDF_g_100g_TS), ADF (ADF_g_100g_TS), ADL (ADL_g_100g_TS), Ca (Ca_UFAG), Mg (Mg_UFAG), and P (P_UFAG) in the feed differed significantly between the control and problem farms. As a result of losses in degrees of freedom, subgroups such as water, soil, blood, and roughage parameters had to be established first, with altitude as a fixed term. From these variables, an overall GLM was created. All of the interactions had to be eliminated due to collinearity. The printout of the final model in STATA with model specifications can be found in Appendix A.

Genomic DNA could be isolated for all yaks and may be analyzed later if candidate genes are recognized or DNA regions of interest are identified using genome-wide association mapping. A pedigree of the yaks was prepared, starting from the animals affected by *urolithiasis*. The analysis of the pedigree did not reveal evidence of a simple Mendelian inheritance of the trait.

## 4. Discussion

### 4.1. The Risk of Urolithiasis to Yak Farming in the Swiss Alps

We demonstrated that *urolithiasis* is a problem affecting yak farming in Switzerland and that there is a connection between an increased number of cases of *urolithiasis* in alpine regions and high calcium and magnesium levels in roughage.

Only seven stones could be analyzed. Most of the cases were lost, since no necropsy was performed or stones were found during slaughter, either at a facility or at a farm.

Yak farming in Switzerland is certainly a niche sector in agriculture. However, the animals thrive quite well despite the environmental factors that vary from their area of origin. Despite their origins from a semi-arid continental area located significantly higher above sea level than found in the Swiss Alps and with a nomadic life that involves a lot of exercise, they have adapted to the temperate climate, lower altitude, and reduced mobility to a certain extent. The lush vegetation, even in the very highest pastures in Switzerland, leads to more frequent cases of greater adipose tissue in yaks when compared to domestic cattle [1]. As a result of the high protein content found in forage at higher altitudes, the freely available mineral content changes. As ampholytes, proteins can bind to minerals differently [6]. This alters the bioavailability of minerals and can affect the risk of *urolithiasis*. In humans, it is generally assumed that there is a genetic predisposition to *urolithiasis*.

The altitudes of the problem farms differed significantly from the control farms. This had an influence on feed composition.

Since almost all cases of *urolithiasis* occurred during winter feeding, it was reasonable to assume that the water uptake, even in the mild winter climate of the Swiss Alps, was low. Additionally, the main food source in winter is hay; in summer, it is grass. Hay has a dry matter content of around 90%, whereas grass has a content of 20% [2,15]. Water availability can be restricted during wintertime due to freezing conditions; however, it was impossible to measure water intake. Inadequate water intake can lead to poorer renal clearance of calcium [6]. It can be considered that the duration of the winter feeding period does not depend solely on altitude. On one hand, some farmers leave their animals on farm pastures for longer around winter and autumn; on the other hand, depending on sun exposure, pastures can also be used earlier in spring.

### 4.2. Urine and Blood

The small number of urine samples did not allow for statistically stable conclusions to be drawn. Five of the six urine samples came from problem farms. The presence of protein in urine can have various causes, and mucoprotein promotes the formation of urinary stones. These parameters would certainly be interesting to investigate and compare in a future study.

Although the total data per parameter were subjected to a normality test and showed a satisfactorily normal distribution, the division into cases and controls showed skewed distributions on several occasions.

Hypercalcemia and hypermagnesemia were detected. In our study, the elevated serum concentrations showed a correlation with elevated Ca and Mg levels in roughage. There was also a correlation with the altitude of the farms [15,16].

### 4.3. Soil Conditions

Soil conditions influence the P value of feed. Fine-grained, clayey raw soils result in the highest P values in feed, and podsols yield the lowest [2]. The roughage from higher altitudes (>1000 m a.s.l.) had a high proportion of herbs, which contain significantly more P than grasses, but, at the same time, significantly more Ca. The concentration in the feed correlated with the values found in the blood. The values for P were below the reference values. The reason for this could be less intensive farming, especially in the case of organic mountain farms. Less intensive fertilization use, as well as the later development stages of the plants during the later prescribed harvest, is decisive here. Although the control farms were also organic farms, they were located in lower regions with different climates and botanical compositions. Only in two of the control farms was the Ca:P ratio close to the target range of 2:1 for ruminant feed. The largest difference in the Ca:P ratio was 8:1, as found in the problem farms [17].

### 4.4. Feed Conditions

The quality of the roughage was good. The samples from the control farms contained only grass-rich roughage. No poisonous plants that may play roles in calcium metabolism were found in the feed samples.

The Ca and Mg values and the degrees of hardness in the water samples were higher in the control farms. Since yaks absorb rather small amounts of water compared to cattle, the amount of minerals in the water probably had a small influence. The amount of water absorbed was not determined.

It was shown that at two locations, in Switzerland and in China, Ca:P ratios between 5:1 and 11:1 occurred at times, and the magnesium values in China were also three times higher in winter than in summer, while the values were similar to our samples [18]. Other major differences are to be found in the protein content and, to a lesser extent, in the crude fiber content of the feed. On average, the roughage used on the problem farms was higher in protein and lower in crude fiber than that of the control farms and also, in some cases, than the samples examined from the area of origin during the winter period.

### 4.5. The Effect of Parasites

In this study, the influence of parasites on the general health of yaks in Switzerland was also investigated to rule out parasites as a possible confounder of *urolithiasis* [19,20,21,22]. The presence of a parasite load did not affect the incidence of *urolithiasis*. Those results will be published elsewhere.

### 4.6. The Effect of Calcium

We can conclude that the high Ca:P ratio and the protein-rich and low-crude fiber feed rations were decisive for the formation of urinary stones for the animals in Switzerland in the comparison of problem farms with control farms. This was also shown in the multivariate analysis, in which altitude, CP, NDF, ADF, and ADL, as well as Ca, Mg, and P values in roughage, significantly influenced the presence of *urolithiasis* in the yaks in Switzerland. Altitude directly or indirectly influences most of the values relating to the feed. Due to the fact that it is probably a multifactorial event, other factors that are not related to feed are involved in the formation of urinary stones.

The influence of 25-Hydroxy-Vitamin-D is unclear. The 25-Hydroxy-Vitamin-D values in the problem farms were, on average, higher than those in the control farms; however, there was no significant difference.

### 4.7. Recommendations

To avoid future cases, yak owners should increasingly have their roughage analyzed in order to compensate for one-sided surpluses or deficiencies in minerals and be aware of water supplementation. The anion gap must be controlled to stabilize pH.

The occurrence of dominant animals hindering the access of other animals to mineral sources has generally been understudied in the case of yaks [3]. Efforts should be made to encourage animals to exercise more during the winter period. Many yaks are obese as a result of good feed quality and large pastures in summer of up to one km^2^ per yak; furthermore, the additional influence of castration is still unclear.

### 4.8. Limitations

Yak farming is niche farming in the Swiss mountains. Most farmers have only some yaks, i.e., one to five. When we selected real yak farms, we wanted to have at least ten yaks per farm. Therefore, we included nearly 100% of the available yak farms in Switzerland. Enlarging the sample with yaks from other countries in the Alps, i.e., Austria, France, and Italy, will raise the heterogeneity. The alpine regions are quite different concerning soil, climate, and husbandry. An international study with a lot more yaks would maybe overcome this issue.

## Figures and Tables

**Figure 1 animals-14-02915-f001:**
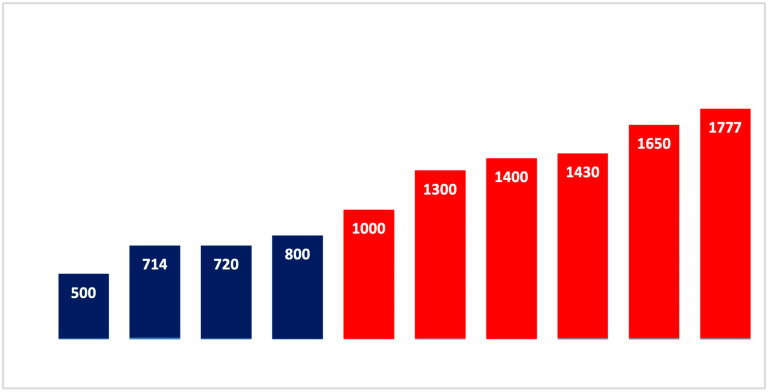
Altitude of farms (m a.s.l.; x-axis: farms; y-axis: altitude): blue = control farms; red = problem farms.

**Figure 2 animals-14-02915-f002:**
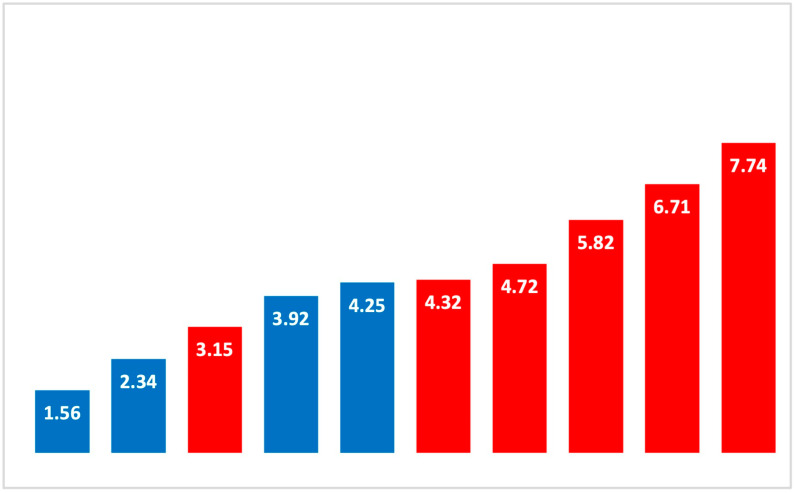
Ca:P ratio in roughage (x-axis: farms; y-axis: Ca:P ratio): red = problem farms; blue = control farms.

**Table 1 animals-14-02915-t001:** Laboratory tests.

Sample	Parameter	Test	Comment
Urine	Na (sodium)	Flame photometer IL 243^®^ instrumentation laboratory, Munich, Germany	Urine previously diluted 1:2 with 1N HCL
Urine	K (potassium)	Flame photometer IL 243^®^ instrumentation laboratory, Munich, Germany	Urine previously diluted to 1:2 with 1N HCL
Urine	Mg (magnesium)	Photometric determination, Cobas Mira S^®^ Roche, Basel, Switzerland	
Serum	FFAs (free fatty acids)	Enzymatic color test, Wako 994-75409 D, Wako Chemicals Europe GmbH, Neuss, Germany	
Serum	BHB (beta-hydroxybutyric acid)	Enzymatic determination, Sigma Diagnostics, Sigma-Aldrich, Basel, Switzerland	Procedure No. 310-UV
Serum	Ca (calcium)	Methylthymol blue reaction, Cobas Mira S^®^ Roche, Basel, Switzerland	
Serum	P (phosphorus)	Phosphomolybdat, UV test, Cobas Mira S^®^ Roche, Basel, Switzerland	
Serum	Mg (magnesium)	Photometric determination using Mg-Kit Bio Mérieux SA, Lyon, France	
Serum	GLDH (glutamate dehydrogenase)		Optimized Standard Method of the German Society for Clinical Chemistry with Oxalglutarate + NADH + NH4
Serum	GGT (gamma-glutamyl-transferase)	Kinetic color test, DGKC, Cobas Mira S^®^ Roche, Basel, Switzerland	L-gamma-glutamyl-3-carboxy-4-nitoanilide + glycilglycerin.
Serum	Hb (hemoglobin)	Photometric measurement of the reaction product cyanmethemoglobin	ADVIA from Siemens, Zurich, Switzerland
Serum	GSH-Px (glutathione peroxidase)	Photometric measurement, AU680 from Beckman Coulter, Krefeld, Germany, with the Randox reagent kit	
Serum	OC (osteocalcin)	Competitive enzyme immunoassay using MicroVue Osteocalcin EIA Kit, Quidel Corporation, Athens, OH, USA	
Serum	PTH (parathormon)	Competitive enzyme immunoassay using MicroVue Human PTH Kit, Quidel Corporation, Athens, OH, USA	
Serum	25-hydroxy-Vitamin-D	Radioimmunoassay using 25-Hydroxy-Vitamin-D RIA, Immunodiagnostic System Farm, Newcastle, UK	

**Table 2 animals-14-02915-t002:** Cadastral zone distribution and sampled yaks.

Zone	Total Farms	Control Farms	Control Yaks	Problem Farms	Problem Yaks
Valley	1	1	10	0	0
Mountain I	1	1	9	0	0
Mountain I–II	1	1	10	0	0
Mountain II–IV	1	1	10	0	0
Mountain III–IV	1	0	0	1	5
Mountain IV	5	0	0	5	50
Total	10	4	39	6	55

**Table 3 animals-14-02915-t003:** Summary of urine analysis (five cases and one control).

Parameter	Mean	SD	Min	p50	Max	N	Benchmarks
Na, mmol/L	15.0	2.0	13.0	14.5	18.0	6	>10
K, mmol/L	292.0	72.0	176.0	299.0	384.0	6	<400
Mg, mmol/L	37.1	9.2	25.2	39.3	49.1	6	>5–10

N: samples; SD: standard deviation; calcium could not be analyzed.

**Table 4 animals-14-02915-t004:** Blood values of yaks on farms with *urolithiasis* and yaks on farms without *urolithiasis*.

Parameter	Case			Control			Difference	Reference
	Mean	SD	N	Mean	SD	N		
Ca, mmol/L	2.5	2.3	56	2.2	0.5	39	***	A: 2.3–2.6 B: 1.48–3.10
P, mmol/L	2.3	2.3	56	2.3	0.6	39	n.s.	A: 1.3–2.4 B: 0.81–4.43
Mg, mmol/L	**1.6**	**1.4**	**56**	**1.3**	0.3	**39**	***	A: 0.80–1.0 B: 0.37–1.00
OC, g/mL	124.2	113.6	56	100.2	110.6	39	n.s.	
PTH, pg/mL	190.7	208.2	47	222.7	381.8	34	n.s.	
25-OH-Vit-D, nmol/L	134.8	127.1	56	114.5	29.0	39	n.s.	
BHB, µmol/L	358.6	332.5	56	284.7	133.9	39	**	A: <900.00
FFA, mEa/L	**0.25**	0.30	**56**	**0.38**	0.20	**39**	***	A: <0.100
GGT, U/L	11.4	16.9	45	23.2	22.9	38	n.s.	A: <25
GLDH, U/L	**38.9**	54.1	**39**	**76.2**	62.9	**55**	***	A: <10

N: samples; SD: standard deviation; n.s. = *p* > 0.05; ** = *p* < 0.01, *** = *p* < 0.001; A: reference of the section for herd health, University of Zurich; B: International Species Information System [3] bold: above reference.

**Table 5 animals-14-02915-t005:** Feed analyses of hay [12,13,14,15].

Parameter	Case			Control			Difference	Reference
	Mean	SD	N	Mean	SD	N		
DM %	94.2	94.8	18	95.7	2.2	10	**	-
CP g/100 g DM	*10.3*	*10.2*	18	*8.4*	*2.4*	10	*****	A: 10.4–14.9
CF g/100 g DM	27.3	27.9	18	**29.5**	2.6	10	***	A: 23.9–27.3
SF g/100 g DM	*1.9*	1.6	18	0.5	*1.2*	10	***	LO: 2.3–3.0
NDF g/100 g DM	**57.5**	58.6	18	**61.4**	3.6	10	***	A: 49.8–55.0
ADF g/ 100 g DM	**35.1**	35.3	18	**36.6**	2.7	10	***	A: 28.2–31.6
ADL g/100 g DM	6.2	5.9	18	5.5	0.5	10	***	N: 3.8–17.1
Ca g/kg DM	**9.1**	**7.9**	18	5.1	**2.4**	10	***	TD: 4.5–6.5
P g/kg DM	*1.5*	1.6	18	*1.7*	0.4	10	n.s.	TD: 3.0–4.0
Mg g/kg DM	2.9	2.5	18	*1.7*	0.8	10	***	TD: 2.0–3.0
K g/kg DM	18.5	20.3	18	22.1	4.5	10	n.s.	U: 15.0–30.0
Na g/kg DM	*0.12*	*0.09*	18	*0.05*	*0.11*	10	n.s.	U: 2.0–3.0
S g/kg DM	*1.7*	*1.7*	18	*1.5*	0.3	10	n.s.	U: 2.0–4.0

N: samples; SD: standard deviation; n.s. = *p* > 0.05; ** = *p* < 0.01, *** = *p* < 0.001; A = average value Agridea 201619; LO = Upper Austrian Chamber of Agriculture [13]; N = Nater S. et al. 200632; U = target area UFAG33; bold: above reference; italics: below reference; TD: Laboratory of the Institute of Animal Nutrition and Dietetics, Vetsuisse Faculty, University of Zurich (www.tierer.uzh.ch; accessed on 3 October 2024); DM: dry matter; CP: crude protein; CF: crude fiber; SF: Soxhlet fat; NDF: neutral detergent fiber; ADF: acid detergent fiber; ADL: acid detergent lignin.

**Table 6 animals-14-02915-t006:** Tap water analyses.

Parameter	Case			Control			Difference
	Mean	SD	*N*	Mean	SD	*N*	
Ca mg/L	44	50.77	6	60	25.11	4	**
Mg mg/L	7	10.32	6	15	9.62	4	***
fH° water hardness	14.62	17.02	6	20.83	9.95	4	***

N: samples; SD: standard deviation; ** = *p* < 0.01, *** = *p* < 0.001.

## Data Availability

The data that support the findings of this study are available from the corresponding author upon reasonable request.

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
