# Peer review of "Urolithiasis* as a Husbandry Risk to Yaks in the Swiss Alps"

_animals, 2024, doi:10.3390/ani14192915_

Round 1

Reviewer 1 Report

Comments and Suggestions for Authors

The manuscript deals with the incidence of urolithiasis in yaks reared in Switzerland, evaluating several soil, feed and animal parameters that could contribute to the formation of urolithiasis.
The study has several limitations, as indicated by the authors. Still, it could be a starting point to begin the evaluation of the species, given the growing interest in breeding this species in Switzerland.
Some suggestions can be found in the attached file.

Regards

Author Response

Dear Reviewer,

Thank you very much for the comments, which help us to further develop the MS.

All spelling errors were of course taken into account and corrected.

why was the age calculated in months and not in years?

Month as a unit instead of Year is more reliable.

explain the differences between the problem and the case

done: Table 3. Blood values of yaks on farms with urolithiasis and yaks on farms without urolithiasis

Michael Hässig

Reviewer 2 Report

Comments and Suggestions for Authors

This study analyzed the potential factors for the formation of urinary calculi in yaks in the Swiss Alps region, and the authors attempted to analyze them from different perspectives and to summarize the main factors that induce urolithiasis. However, the writing in the manuscript is not rigorous enough and the experimental design is not reasonable enough.

1. The sample is too small, when the urine was collected, the authors only collected a control group. Whereas for statistics, when the group contains more than 3, it is more valuable for statistics. In addition, in line 228, the authors wrote there were 2 control groups, the grouping data is inconsistent.

2. Lines 278-279, there is an obvious error: the control farms show significantly higher values than the control farms, there were two “Control farms”.

3. When the authors analyze the multifactorial, the statistical methods should be described in more detail, for example, in the linear model, how each parameter is designed should be listed in more detail.

4. Water quality and water intake are important factors in causing urolithiasis, and the authors did not seem to measure differences in water intake in their experiments.

5. There are too many factors analyzed in the whole experiment, and when analyzing different factors, there may be a possibility of mutual interference between the factors, the authors should consider using a generalized linear mixed model to analyze the effects of different factors. Combine the overall effects of these factors.

Comments on the Quality of English Language

The quality of English language is fine.

Author Response

Dear Reviewer,

Thank you very much for the comments, which help us to further develop the MS.

  1. The sample is too small, when the urine was collected, the authors only collected a control group. Whereas for statistics, when the group contains more than 3, it is more valuable for statistics. In addition, in line 228, the authors wrote there were 2 control groups, the grouping data is inconsistent.

Correct: There was no analytical statistics done with urine samples.

Table 2. Summary of urine analysis (five cases and one control); reference in text skipped.

  1. Lines 278-279, there is an obvious error: the control farms show significantly higher values than the control farms, there were two “Control farms”.

We cannot see any problem in this part:

For the minerals of the water and the degree of hardness, the control farms show significantly higher values than the control farms (Table 5). The water is softer in the control farms than in the problem farms.

Table 5. Tap Water analyses.

Parameter

Case

Control

difference

Mean

SD

N

Mean

SD

N

Ca mg/l

44

50.77

6

60

25.11

4

**

Mg mg/l

7

10.32

6

15

9.62

4

***

fH° water hardness

14.62

17.02

6

20.83

9.95

4

***

N: samples; SD: standard deviation; ** = P < 0.01, *** = P < 0.001.

  1. When the authors analyze the multifactorial, the statistical methods should be described in more detail, for example, in the linear model, how each parameter is designed should be listed in more detail.

In M&M added: The printout of the final model in STATA with model specifications is given in attachment 1.

In results GLM was rewritten and the STATA printout was added with all information to the model in the attachment 1:

By means of a stepback procedure taking into account the AIC (Akaike information criterion), the multivariate model (GLM, general linear model) with variance function Gaussian, shows that the altitude (MüM), crude protein (RP_g_100g_TS), NDF (NDF_g_100g_TS), ADF (ADF_g_100g_TS), ADL (ADL_g_100g_TS), Ca (Ca_UFAG), Mg (Mg_UFAG), and P (P_UFAG) in the feed differ significantly in the control and problem farms. As a result of the loss of degrees of freedom, subgroups such as water-, soil-, blood- and roughage-parameters had to be set up first with altitude as a fixed term. From these variables, an overall model for GLM was created. All interactions had to be eliminated due to colinearity. The printout of the final model in STATA with model specifications can be found in attachment 1.

Attachment 1: STATA printout of GLM

  1. stands for unary operator to treat as continuous and Problembetrieb for cases or controls:

glm Problembetrieb c.MüM  c.RP__g_100g_TS c.NDF_g_100g_T c.ADF_g_100g_TS c.ADL_____g_100g _TS c.Ca_UFA c.Mg_UFAG c.P_UFAG

Iteration 0:   log likelihood =  93.020057

Generalized linear models                         Number of obs   =        104

Optimization     : ML                             Residual df     =         95

                                                  Scale parameter =   .0107139

Deviance         =  1.017823555                   (1/df) Deviance =   .0107139

Pearson          =  1.017823555                   (1/df) Pearson  =   .0107139

Variance function: V(u) = 1                       [Gaussian]

Link function    : g(u) = u                       [Identity]

                                                  AIC             =   -1.61577

Log likelihood   =  93.02005725                   BIC             =  -440.1993

-------------------------------------------------------------------------------

                  |                 OIM

   Problembetrieb | Coefficient  std. err.      z    P>|z|     [95% conf. interval]

------------------+----------------------------------------------------------------

              MüM |   .0006504   .0000433    15.04   0.000     .0005657    .0007352

    RP__g_100g_TS |   .4471499   .0408438    10.95   0.000     .3670974    .5272023

    NDF_g_100g_TS |  -.2243896   .0163061   -13.76   0.000     -.256349   -.1924302

    ADF_g_100g_TS |   .6690923   .0613285    10.91   0.000     .5488906     .789294

ADL_____g_100g_TS |   .7555633   .0658497    11.47   0.000     .6265002    .8846265

          Ca_UFAG |   .2636711   .0255685    10.31   0.000     .2135578    .3137844

          Mg_UFAG |  -1.390118   .1236952   -11.24   0.000    -1.632556    -1.14768

           P_UFAG |   1.081129   .1270552     8.51   0.000     .8321055    1.330153

            _cons |  -20.62702   2.288495    -9.01   0.000    -25.11239   -16.14165

-----------------------------------------------------------------------------------

  1. Water quality and water intake are important factors in causing urolithiasis, and the authors did not seem to measure differences in water intake in their experiments.

We also believe that water quality and water intake are important factors in causing urolithiasis, but it was impossible to measure water intake during grazing season in the mountains. We believe that water intake is not a very important factor for urolithiasis in our setting due to the fact that Swiss Alps are quite humid compared to Central Asia. Water quality was tested for as written in the manuscript.

The following part was rewritten:

Water-availability can be restricted I wintertime due to freezing conditions, but it was impossible to measure water intake. Failure of adequate water intake can lead to poorer renal clearance for calcium.

  1. There are too many factors analyzed in the whole experiment, and when analyzing different factors, there may be a possibility of mutual interference between the factors, the authors should consider using a generalized linear mixed model to analyze the effects of different factors. Combine the overall effects of these factors.

Done

Michael Hässig

Reviewer 3 Report

Comments and Suggestions for Authors

In general, the article seems very interesting and innovative, so I congratulate the authors. Although the sample is relatively small, the type of analyses carried out is very comprehensive. Despite the positive points, some questions clearly need to be clarified.

Lines 80-81: How is a ‘problem farm’ defined? Just by the presence of urolithiasis cases in the last 11 years? How many cases are needed to be considered a ‘problem farm’? Even if only a single case appeared about 10 years ago, is that farm still classified as a ‘problem farm’? Two farms went from being control farms to problem farms because new cases of urolithiasis appeared during the study period? Clarify.

Lines 91-93: Was this farm that imported feed from Germany classified as a control farm or a problem farm?

Table 1: It is important that the word ‘control’ overlaps with the words ‘Farms’ and ‘Yaks’. This way the table looks a bit confusing. The same goes for the word ‘Problem’.

Lines 206-209: They should objectively define what ‘high altitude’, ‘very high altitude’ and ‘cadrastal zones’ are. They should also indicate which other factors have been taken into account in each zone. This information should appear in the material and methods chapter and not in the results.

Discussion: It seems to me that there is a lot of information that the authors of this paper are not responsible for (and that it is not possible to conclude from the information obtained here alone) and that some authors may have forgotten to cite. Please confirm that all the data that is not original has been properly cited. For example, where you got the information in lines 305 to 310?

Lines 315-316: Does this information result from the laboratory analysis carried out in this study? If so, it should be clarified. If not, the source of the information should be indicated.

Author Response

Dear Reviewer,

Thank you very much for the comments, which help us to further develop the MS.

Lines 80-81: How is a ‘problem farm’ defined? Just by the presence of urolithiasis cases in the last 11 years? How many cases are needed to be considered a ‘problem farm’? Even if only a single case appeared about 10 years ago, is that farm still classified as a ‘problem farm’? Two farms went from being control farms to problem farms because new cases of urolithiasis appeared during the study period? Clarify.

The definition of cases and controls is rewritten: One case of Urolithiasis within a farm was considered as a case. Initially, 4 problem farms, i.e., farms in which urolithiasis in the last 11 years had been detected (cases) and 6 farms without urolithiasis in the last 11 years were included in the study (controls). During the study, two farms switched from control farms to problem farms (6 problem farms and 4 control farms) due to the extended anamnesis, because new cases of Urolithiasis appeared during the study period.

Lines 91-93: Was this farm that imported feed from Germany classified as a control farm or a problem farm?

Inserted: (a control farm)

Table 1: It is important that the word ‘control’ overlaps with the words ‘Farms’ and ‘Yaks’. This way the table looks a bit confusing. The same goes for the word ‘Problem’.

Done

Lines 206-209: They should objectively define what ‘high altitude’, ‘very high altitude’ and ‘cadrastal zones’ are. They should also indicate which other factors have been taken into account in each zone. This information should appear in the material and methods chapter and not in the results.

This information is transferred to Material and was rewritten:

The problem farms are located exclusively in mountain zones III (farms on high altitude and other factors) and IV (farms on very high altitude and other factors), the control farms in all cadastral zones. The catastral zones are defined by the Swiss Federal Office of Agriculture, The delimitation criteria are climate, traffic situation, altitude and exposure. The most important criteria is altitude. The altitude of each farm is given.

Discussion: It seems to me that there is a lot of information that the authors of this paper are not responsible for (and that it is not possible to conclude from the information obtained here alone) and that some authors may have forgotten to cite. Please confirm that all the data that is not original has been properly cited. For example, where you got the information in lines 305 to 310?

We confirm that all the data that is not original has been properly cited. As written in Methods there were several different labs involved in this study.

The information in lines 305 to 310 was collected during the anamnestic procedure before taking blood and urine samples.

Lines 315-316: Does this information result from the laboratory analysis carried out in this study? If so, it should be clarified. If not, the source of the information should be indicated.

Source is added: [22].

Michael Hässig

Reviewer 4 Report

Comments and Suggestions for Authors

The article is well designed and written. The results are well presented and correctly discussed. Therefore, it represents a scientific contribution.

Just some suggestions and questions.

What were the symptoms exhibited by the animals and what were the complaints presented by the farmers?

What factors led the researchers to carry out this study?

Cite this information in the text

Author Response

Dear Reviewer,

Thank you very much for the comments, which help us to further develop the MS.

What were the symptoms exhibited by the animals and what were the complaints presented by the farmers?

Added: The complaints of the yaks presented by the farmers were severe colic, depression and anorexia.

What factors led the researchers to carry out this study?

Added in Introduction: The Department of Farm Animals, University of Zurich, Switzerland, was asked to help in 10 cases of urolithiasis in yaks during 2006 and 2014. Calcium carbonate uroliths were found and a 6-fold calcium overhang in roughage in an affected farm was shown.

Michael Hässig

Round 2

Reviewer 2 Report

Comments and Suggestions for Authors

The explanation is not Insufficient.